

# Construct and evaluate the classification models of six types of geological hazards in Bijie city, Guizhou province,China

JieQing Shi[1], Jin Zhang[1], and ChaoYong Shen[2]

[1]Taiyuan University of Technology, Taiyuan 030024,China
[2]The Third Surveying and Mapping Institute of Guizhou Province, Guiyang 550004, China

**Correspondence:** JinZhang (jinzhanggis@126.com)

**Abstract.** Debris flow, landslide, ground collapse, collapse and ground collapse are the dominating geological hazards in Bijie city, Guizhou province, which is situated in the area with high natural hazards in China. The primary purpose of this study is to construct different classification models by using the disaster conditioning factors of geological hazards and to evaluate the performance of the models in the classification of geological hazards in Bijie city. At the same time, the nonlinear

relationship between various geological hazards and conditioning factors will be discussed. Firstly, the manual field survey data of Bijie city in 2019 were applied to construct and draw inventory map of six geological hazards. Then 16 conditioning factors were established from various data sources. According to the ratio of 70:30, the geological hazard location points were randomly divided into the training and validation set to complete the training and verification process of the classification models. In order to select the optimal subset of the conditioning factors, the multicollinearity of these factors was assessed

using tolerances and variance inflation factors(VIF) and Pearson's correlation coefficient, and factors with multicollinearity were excluded to optimize the model. Subsequently, ten classification models were structured, and the models were verified and compared by using the receiver operating characteristic(ROC), precision, sensitivity, Kappa coefficient and F1 values.In addition, the Friedman test was used to identify statistically significant differences between the results of the classification model used in this research. In general, average Area Under Curve (AUC) values under the ROC curves of the 10 classification

models is above 0.8, indicating that all models have a corresponding high prediction ability. Among them, the average AUC value(0.941), AUC values for individual geological hazards (collapse: 0.949, ground crack: 0.907, ground collapse: 0.952, landslide: 0.830, displacement flow: 0.963, slope: 0.922),Kappa coefficient (0.845), Macro F1(0.851) and Micro F1(0.878) of SVM all had the highest values.

## 1 Introduction

Most cities in China are located in regions that are extremely vulnerable to a wide range of natural hazards,particularly geological hazards, and the assessments for hazardousness, vulnerability and risk of natural hazards for China shows that on the whole, high hazardousness regions concentrate in western (Wang et al., (2008), Liu et al., (2012),Zhuang et al., (2016)). Guizhou province, located in the plateau region of southwest China, not only has extensive carbonate rock distribution (70%) and karst development; In addition, due to the large crustal uplift and severe deformation recently, as well as the induction



of human activities such as mining, geological disasters occur frequently in Guizhou(Cao and Wang, (1993)). Among them, debris flow, landslide, unstable slope, ground collapse, collapse and ground crack are the main geological disasters in Guizhou.

   Debris flows are fast-moving, high-density slurry of water, sediments and debris that travels under gravity and are endued with enormous destructive power (Cannon et al., (2007);Cannon et al., (2010); Staley et al., (2017)). And the debris flows in Guizhou province are mainly distributed in the western part of the province ,ranging from several hundred thousand to several

million in size.While landslide refers to the deformation and failure phenomenon of the slope caused by water or increased load, or by air, or by other actions caused by part of the ground body sliding down a weak face(Malamud et al., (2004)).Among all kinds of geological hazards in Guizhou province, landslide is the most dangerous one, especially in the western part of the province, followed by the eastern part. Collapse is also a very common dynamic geological phenomenon of slope instability in Guizhou, and it mostly occurs in deep canyons or steep areas. In addition to the slopes that have already slipped, there are also

many unstable slopes in Guizhou Province that do not slide but may have potential slip surfaces.As for ground collapses and cracks, most of the ground collapses areas that have been found in the province occur in carbonate areas. Guizhou's unique types of karst landforms, coupled with human activities such as groundwater mining, mining, and engineering water storage are the main causes of surface collapse in the province. Ground cracks can be divided into natural and artificial according to their causes. Most naturally formed ground cracks are tensile cracks caused by slope instability, with a length of several

meters, a depth of several meters, and small scales. The ground cracks caused by human activities, such as the scale of the ground cracks in Guiyang and Anshun, Guiyang city, etc., has a large direct or indirect economic loss of hundreds of thousands of yuan. But in fact, no matter what kind of geological disaster, they will seriously endanger people's lives and property safety, public facilities and ecological environmentWang and Yang, (2008); Zhang et al., (2019)).

   From the traditional discriminant analysis and logistic regression to classification branch of machine learning, classification

has always been aimed at using marked information to discover classification rules and construct classification model, so as to output data attribute characteristics without marked information (Williams et al., (2006)). As an important part of machine learning besides regression and clustering analysis, the classification method can be divided into single classification algorithm and integrated classification algorithm according to whether the classifier is single or not. Multi-layer perceptron neural network, naïve bayes, k-nearest neighbor, decision tree and support vector machine all belong to common single classification

algorithms(Zaslavsky, (2009)).With the increasingly prominent problems such as complex data structure, large data volume and uneven data quality, Hansen and Salamon proposed a new machine learning method – integrated learning, which integrates the predicted results of several base classifiers in a certain way or rule, thus effectively overcoming overlearning and improving the classification effect (Hansen and Salamon, (1990)). It is the main idea of two Boosting algorithms first proposed by Schapire and Freund to train the base classifier iteratively by the method of heavy weight, and then to combine the base classifier by the

method of sequential linear weighting (Schapire, (1990); Freund, (1997)). Nevertheless, Boosting algorithm requires knowing the lower limit of classification accuracy of weak classification algorithm in advance, but it is difficult to determine in practice. Based on Boosting idea, Freund further proposed AdaBoost algorithm (Freund and Schapire, (1995)). And the stochastic forest algorithm proposed by Breiman in 2001 focuses on the integrated learning of decision trees (Breiman, (2001)). In the face of such a variety of classification algorithms and models, it is necessary for us to select the appropriate classification algorithm in



practice by combining the characteristics of data sets and research areas. Therefore, in this study, we should establish , analyze and compare the performance of different classification models in the research area based on the disaster conditioning factors of geological hazards, so as to obtain the classification model applicable to the classification of geological hazards in Bijie area. On this basis, we will also study the nonlinear relationship between all kinds of geological hazards and conditioning factors in research area.

This paper is mainly composed of the following parts. The first part mainly elaborates the theoretical research foundation, research purpose and main research content of this paper. The second part describes the general situation of the research area and its resources and environment. The third part recommends the classification methods and theories involved in this research. The fourth part is the data preparation stage, which introduces the data used in the research in detail. The fifth part is the conclusion and analysis of this research. The discussion and conclusion are located in the sixth part of the article.

## 2 Study area

Bijie city is located in the northwest of Guizhou province, where geological disasters are relatively serious. It lies between longitude 105°36 '– 106°43' and latitude 26°21 '– 27°46'(Fig1). The total area of the city is nearly 26,900 square kilometers, and the highest elevation in the territory is 2,900.6 meters, which is also the highest point in Guizhou and the lowest elevation is 457 meters.The population of the study area is 70.298 million people, of whom 25.88% population belong to minority. Bijie
belongs to the humid climate of the northern subtropical monsoon, with abundant rainfall. The average annual rainfall is 854-1444mm,while average annual sunshine and annual temperature were between 1140h and 1450h and 13.2° respectively.There are 37 kinds of proven mineral resources, of which coal occupies a dominant position, with a distribution area of 4,000 km2, occupying 45.9% of Guizhou's coal resources(Wang et al., (2015)). Near-horizontal (dip angle below 8°) and gently inclined (dip angle between 8° and 25°) coal seams are the main coal seam outputs in Bijie.The basin area of the Yangtze River Basin
within the city is 25,600 km2, and the basin area of the Pearl River Basin is 1,239 km2, accounting for 95.39% and 4.61% of the city's total area, respectively. Structurally, Bijie has a complex geological environment due to extensive faults(Yin et al., (2016)), which is dominated by karst topography and mountains and hills. The terrain in the area is high in the west and low in the east. The main structural changes that control the tectono-stratigraphic framework of Bijie and its adjacent areas are the Duyun, Guangxi, Central Indosinian, Late Yanshanian, and Himalayan movements. The outcrops are mainly sedimentary
rocks, less magmatic rocks. In sedimentary rocks, the majority of them are carbonates. Next is the coal measure sand shale, then is the purple sand shale and the purple sand mudstone, finally is the argillaceous rock kind.

**Figure 1.** Geological hazards inventory map and location of the study area





## 3 Methodology

### 3.1 Factors analysis

There may be a high correlation between the conditioning factors in the initial data set, which leads to the wrong systematic

analysis, so we adopted multicollinearity analysis in statistics to solve this problem (Dormann et al., (2013)). The variance decomposition proportions (Schuerman, (1983)), the conditional index (Belsley,1991), Pearson's correlation coefficients (Booth et al., (1994)), and and the variance inflation factors (VIF) and tolerances (Hair, (2009);Dan and Richard, (2012)) are both multicollinearity quantization methods. Among them, VIF and tolerances are often used to check the multicollinearity of conditioning factors in disaster research (Bui et al., (2011)), while Pearson's correlation coefficients method is widely used in

various fields (Dormann et al., (2013)).

In order to check the multicollinearity, the VIF and tolerances measure the variation in the standard errors of the conditioning factors; thus, the higher the standard errors, the greater the multicollinearity (Allison, (1999)). The VIF is greater than 10 or the tolerance is less than 0.1 indicating a potential multicollinearity problem in the data set (Hair, (2009); Keith, (2006)). Pearson's correlation coefficients method was used to evaluate the correlation coefficient of the two conditioning factors, and its formula

was defined as formula 1. Pearson's correlation values> 0.7 dicates a high collinearity between the two factors (Booth et al., (1994)).

$$\rho_{X,Y} = \frac{cov(X,Y)}{\sigma_X \sigma_Y} = \frac{E(XY) - E(X)E(Y)}{\sqrt{E(X^2) - E(X^2)}\sqrt{E(Y^2) - E(Y^2)}} \tag{1}$$

Where $cov(X,Y)$ is expressed as the covariance of the conditioning factors $X$ and $Y$, and $\sigma_X$ are $\sigma_Y$ the standard deviation of $X$ and $Y$, respectively.

### 3.2 Classification model

#### 3.2.1 Logistic regression(LR)

Logistic regression can reveal the relationship between the target variables and multiple prediction variables and predict the occurrence probability of an event(Cox, (1959)). On the basis of the linear regression model, LR regression uses sigmoid function to compress the results of the linear model to between [0,1], so that it has probability significance. In the statistical

analysis of LR, the predictive variables can be continuous or discrete and do not need to satisfy the normal distribution.The formula of LR is as follows:

$$y = \frac{1}{1 + e^{-(\alpha + \beta_1 x_1 + \beta_2 x_2 + ... + \beta_n x_n)}} \tag{2}$$

Where $\alpha$ is a constant, $n$ is the number of independent variables, $x_i(i = 1, 2, ..., n)$ is the predictor variables and $\beta_i(i = 1, 2, ..., n)$is the coefficient of the LR.




### 3.2.2 Linear discriminant analysis(LDA)

Linear discriminant analysis is required to achieve low coupling between classes and high aggregation degree within classes, that is, the values of the intra-class dispersion matrix are smaller and the values of the inter-class dispersion matrix are larger(Fisher, (1936); Mclachlan, (2004)). Suppose an n-dimensional space has m samples and c classifications, respectively expressed as $x_1, x_2, .., x_m$, $n_i$ represents the number of samples belonging to class i. LDA needs to select the n-dimensional column vector $\varphi$ that maximizes $J_{fisher}(\varphi)$ as the projection direction. The formula of $J_{fisher}(\varphi)$ is as follows:

$$J_{fisher}(\varphi) = \frac{\sum_{i=1}^{c} n_i \varpi^T (\mu_i - \mu)(\mu_i - \mu)^T \varphi}{\sum_{i=1}^{c} \sum_{x \in \omega} \varpi^T (x - \mu_i)(x - \mu_i)^T \varphi} \tag{3}$$

Where $\mu$ is the mean of all samples and $\mu_i$ is the mean of samples of class $i$.

### 3.2.3 Naïve Bayes Classifier(NBC)

Naïve Bayes Classifier assumes that the influence of the value of any attribute on a known class is independent of the values of the remaining attributes (Maron and Kuhns, (1960)). The purpose is to use the joint distribution of feature output Y and feature X to solve the posterior probability, and based on the value of posterior probability for classification (Han et al., (2011)). For a sample set D with m samples, n features, and $C_k$ sample categories:
$D = \{(x_1^{(1)}, x_2^{(1)}, ..., x_n^{(1)}, y_1), (x_1^{(2)}, x_2^{(2)}, ..., x_n^{(2)}, y_2), ...(x_1^{(m)}, x_2^{(m)}, ..., x_n^{(m)}, y_m)\}$.
The NBC can be expressed as:

$$P(Y = C_k|X = x) = argmax_{C_k} \frac{P(Y = C_k)\prod_{j=1}^{n} P(X_j = x_j|Y = C_k)}{\sum_k P(Y = C_k)\prod_{j=1}^{n} P(X_j = x_j|Y = C_k)} \tag{4}$$

Among them, $P = (Y = C_k)$ is expressed as a prior probability, $P(X_j = x_j|Y = C_k)$ is expressed as a conditional probability, and the denominator is expressed as a full probability according to the concept, so it is the same for all Ck. The above-mentioned posterior probability can be further expressed as:

$$P(Y = C_k|X = x) = argmax_{C_k} P = (Y = C_k)\prod_{j=1}^{n} P(X_j = x_j|Y = C_k) \tag{5}$$

### 3.2.4 Multi-layer perceptron (MLP)neural network

Multi-layer perceptron neural network is one of the most effective artificial neural networks. It consists of an input layer, one or more linear threshold units called hidden layers, and a final layer linear threshold unit called output layer. Its basic structure is shown in Fig.2. Each layer except the output layer includes biased neurons and is fully connected to the next layer. For each training instance, the MLP neural network algorithm first performs forward prediction and obtains the measurement error, then traverses each layer in reverse to measure the error contribution of each connection, and finally adjusts the connector weights slightly to reduce the error(Haykin, (1998);Kavzoglu and Mather, (2003)).


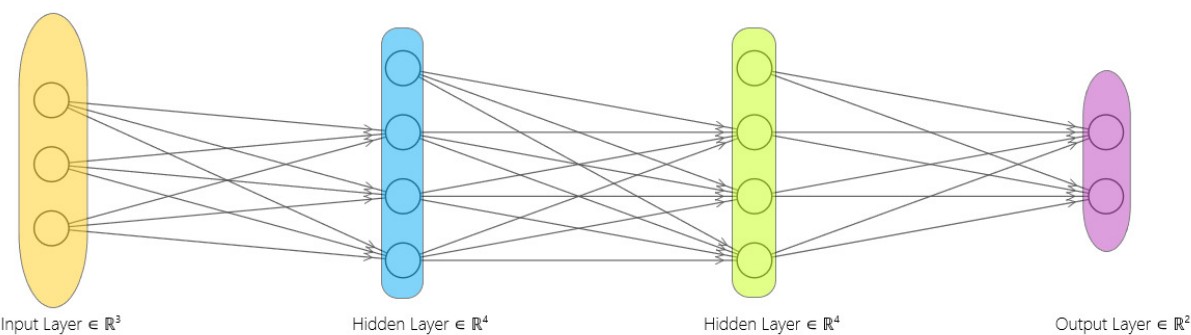

**Figure 2.** The basic structure of MLP neural network

The weight initialization of the MLP neural network not only affects the final convergence result and convergence speed, but also avoids the problems of gradient vanishing and gradient explosion (Salakhutdinov and Hinton, (2006)). There are many methods for weight initialization. In this study, we used the Xavier initialization method (Glorot and Bengio, (2010)), which
initializes the parameters to a uniform distribution in the following range:

$$[-\sqrt{\frac{6}{n^k + n^{k+1}}}, \sqrt{\frac{6}{n^k + n^{k+1}}}] \qquad (6)$$

Where $n^k$ and $n^{k+1}$ are represented as the number of neurons in the input layer and the output layer

### 3.2.5 Support vector machine(SVM)

The original support vector machine is a type of binary classifier, whose basic model is a linear classifier defined as spacing
maximization on the feature space. It can also be used as a nonlinear classifier to solve the classification problem of nonlinear data sets by introducing a kernel function (Vapnik, (1998)). In order to construct support vector machine classifiers suitable for multi-classification, there are two mainstream methods: the first method is to construct multiple type two classifiers and combine them to achieve multi-class classification, such as one- against –rest(Bottou et al., (1994)), one- against –one(Knerr et al., (1990)), directed acyclic graph SVM(DAGSVM) (Platt et al., (2000)).The second method is to directly consider the
parameter optimization of all sub-classifiers simultaneously in an optimization formula (Weston and Watkins, (2005)). In this paper, one-against-one method is adopted (Hsu and Lin, (2002)). The basic idea is to construct a second-class SVM classifier for any class of the training set, that is, for an M-class problem, we need to construct a $2^{-1}M(M-1)$ sub-classifier. When testing, all sub-classifiers are used to process all test data, and the category with the most votes is the category of test data.




### 3.2.6 Decision Tree(DT)

The learning of decision tree adopts a top-down recursive method. Its basic idea is to construct a tree with the fastest entropy decline based on the measure of information entropy. The entropy value is 0 at the leaf node, and its basic structure is shown in Fig.3(Xie and Liu, (2010)). At present, there are three algorithms of decision tree model: ID3,C4.5(Quinlan, (1993)) and CART(Breiman, (2001)). In this study, C4.5 algorithm was selected for classification. The information gain measurement adopted by ID3 has an inherent bias, which preferentially selects the feature with more attribute values, owing to a relatively

large information gain. In order to avoid this deficiency, the gain ratio is used as the criterion for selecting branches in C4.5. The information gain ratio will penalize the feature with more values by introducing an item called split information. The split information of calculation feature A to data set D is expressed in formula 1, and the information gain ratio is expressed in formula 2. In addition, C4.5 compensates for the inability of ID3 to handle the continuum of feature attribute values.

$$SplitInformation(D, A) = -\sum_{i=1}^{n} \frac{|D_i|}{|D|} log \frac{|D_i|}{|D|} \tag{7}$$


$$GainRatio(D, A) = \frac{g(D, A)}{SplitInformation(D, A)} \tag{8}$$

Where $n$ is the number of values of feature $A$, $D_i$ is the number of sub-samples with the same value of feature $A$, and $g(D, A)$ is the information gain

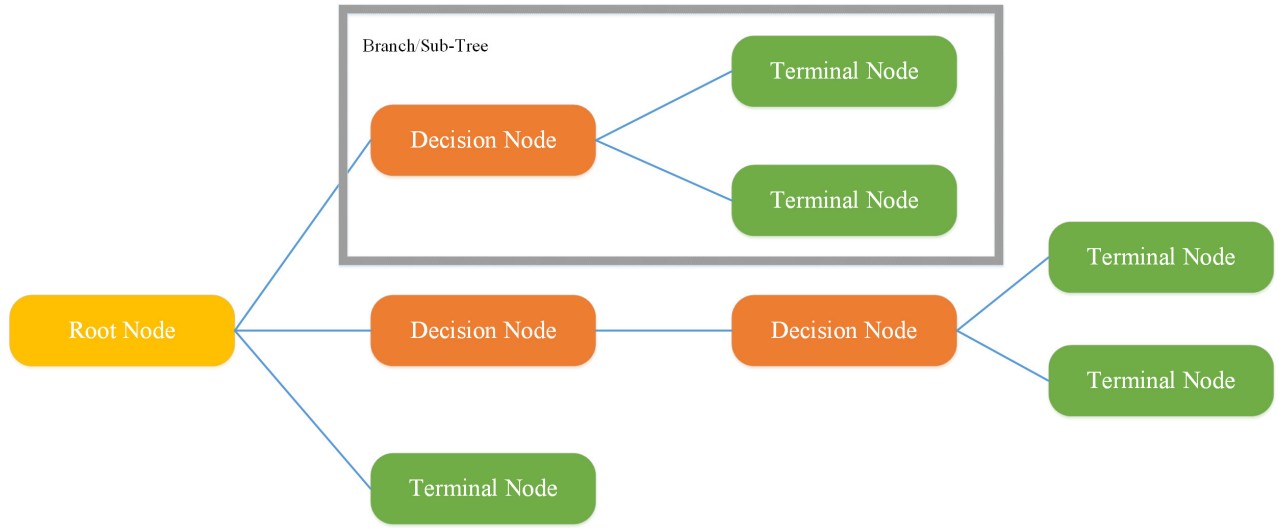

**Figure 3.** The basic structure of DT

### 3.2.7 K-nearest neighbor(KNN)

Among the existing classification methods, K-nearest neighbor classification is a simple, effective and nonparametric method(Cover and Hart, (2003)). The overall idea is to make D as the training data set. When test set D appears, compare D with all samples




in D and calculate the similarity (or distance) between them. Select the most similar samples in the first k from D, and the category of D is determined by the category with the most occurrence in the samples of k closest neighbors. The key part of the k-nearest neighbor algorithm is the distance function. The familiar measurement methods of distance are Euclidean distance, cosine, correlation and Manhattan distance etc. In the above common distance measurement methods, the difference between different characteristics of the training data set and the test set is treated as the same by default. In practice, this approach often fails to meet the requirements. For example, in this study, the measuring unit of altitude is meters and the measuring unit of slope is degree. If these two attributes are treated equally in the calculation of distance, the judgment will be wrong because of the difference between the measurement standards of different characteristics. For this reason, in this study, we adopt the Mahalanobis distance, which is not affected by the dimension, and it is proposed by P. C. Mahalanobis to represent a measurement method of data covariance distance. It is an effective method to calculate the similarity of two unknown sample sets. Different from Euclidean distance, it takes into account the relation between various characteristics and is scale independent. The purpose of Mahalanobis distance is to normalize the variance, so as to make the relation between features more consistent with the actual situation. The formula is as follows:

$$D_M(x) = \sqrt{(x-\mu)^T S^{-1} (x-\mu)} \tag{9}$$

Among them, $x = (x_1, x_2, x_3, ..., x_p)^T$ is an element in the X sample with p variables, and $\mu = (\mu_1, \mu_2, \mu_3, ..., \mu_p)^T$ is expressed as the mean.

### 3.2.8 Adaptive Boosting(Adaboost)

Adaptive Boosting is the most famous representative of Boosting algorithms(Freund and Schapire, (1995)). It changes the probability distribution of the data by increasing the weight of the samples misclassified by the previous weak classifier and reducing the weight of the samples correctly classified. In this way, the data that are not properly classified will receive more attention from the weak classifier in the next round due to its increased weight. In addition, Adaboost will increase the weight of the weak classifier with small error rate and make it play a bigger role in the voting. However, the weight of the weak classifier with large error rate is reduced, so that it plays a small role in the voting. The specific implementation process of Adaboost adopts the idea of iteration. Each iteration only trains a weak classifier, and the trained weak classifier will participate in the use of the next iteration (Fig.4). From right to left, you can see the final sum and the sign function, and before you see the left sum, the dotted line in the graph shows the iterations of the different rounds. Each iteration will train the Weak Classifier(i) through the data set data and data weight W(i), and obtain its classification error rate, so as to calculate its Weak Classifier weight alpha(i). Then, through the method of weighted voting, all weak classifiers are allowed to conduct weighted voting to get the final prediction output, and the final classification error rate will be calculated. Finally, if the final error rate is lower than the set threshold (such as 5%), the iteration ends. If the final error rate is higher than the set threshold, the data weight will be updated as $W(i+1)$.




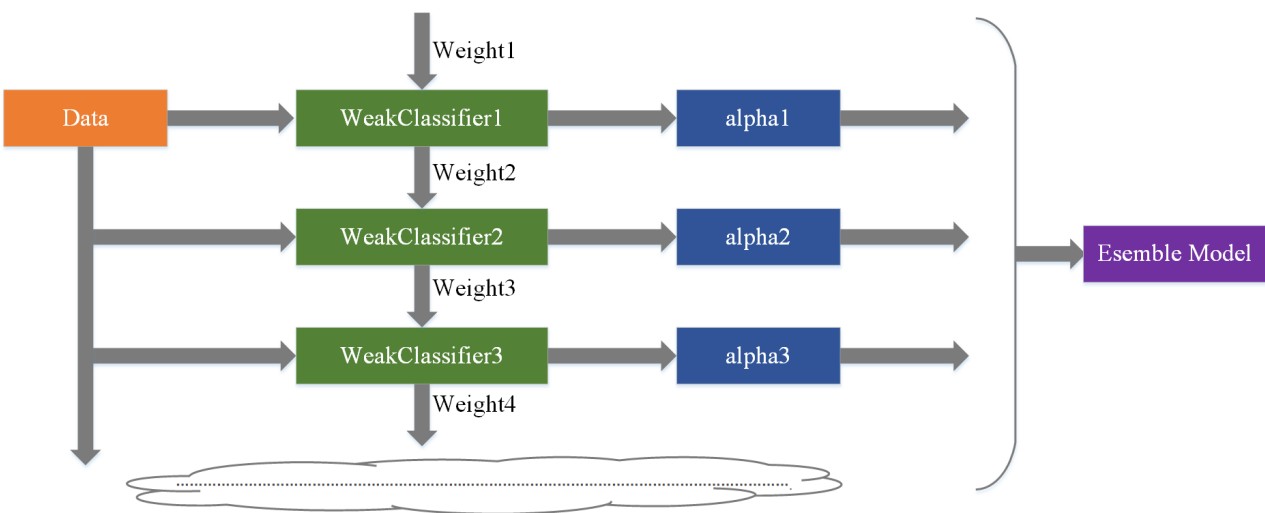

**Figure 4.** The basic structure of Adaboost

### 3.2.9 Gradient Boosting Decision Tree(GBDT)

Like Adaboost, Gradient Boosting Decision Tree is a process of repeatedly selecting a model with general performance and
210 adjusting the model based on the performance of the previous model. The difference is that Adaboost locates the deficiency
of the model by increasing the weight of the wrong fraction data, while Gradient Boosting locates the deficiency of the model
by calculating the Gradient(Ma et al., (2017)). Therefore, compared with Adaboost, GBDT can use more kinds of objective
functions, and when the objective function is the mean square error, the negative gradient value of the loss function calculated
in the current model is the residual. The training process of GBDT is shown in Fig.5. The model training process can be
215 described as: the model trains M rounds in total, and each round produces a weak classifier. The loss function of the weak
classifier is the current model. GBDT utilizes the value of the negative gradient of the loss function in the current model
as the approximate value of the residual in the algorithm of the regression problem promotion tree to fit a regression tree to
determine the parameters of the next weak classifier. Specific to the choice of loss function itself, there are square loss function,
logarithmic likelihood loss function and exponential loss function, etc. In this study, the logarithmic likelihood function is
220 selected as the loss function because it has better optimization for multivariate classification.





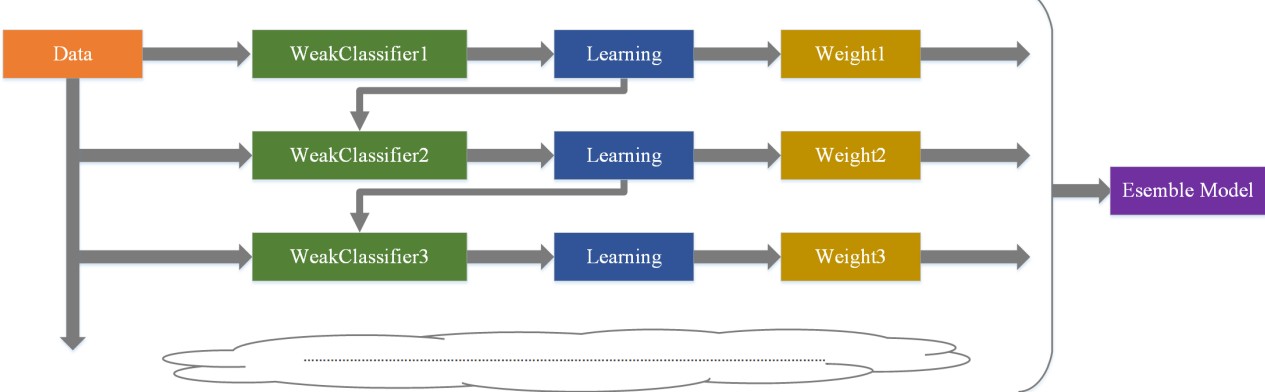

**Figure 5.** The basic structure of GDBT

### 3.2.10 RandomForest(RF)

Random forest is an algorithm that integrates multiple trees with the idea of integrated learning(Breiman, (2001)). Its basic unit is the decision tree, and the output category is determined by the mode of the output category of the individual decision tree.In a random forest, random is the core, and its randomness has two meanings. The first is that it will randomly put back in the original training data and select the same amount of data as the training sample. The second is to establish a decision tree by selecting some features from the random selection features.These two kinds of randomness make the correlation between each decision tree small and further improve the accuracy of the model.

## 4 Data preparation

### 4.1 Geological hazards inventory

In this study, we constructed a detailed and reliable inventory of six major geological hazards in Bijie area based on the manual field survey in 2019 (Fig.1), as shown in table 1. Landslide accounts for the largest proportion (29.56%) of the six geological hazards, which is consistent with the situation of the whole province of Guizhou. This is followed by collapses and unstable slopes, which account for 25.16% and 20.56% of all, respectively. Compared with other geological disasters, the proportion of debris flow and ground cracks is much smaller, accounting for only 4.11% and 5.71% of the total. No matter what kind of disaster, the scale of the disaster varies greatly. Take the collapses as an example, the largest collapse is six orders of magnitude different from the smallest collapse. The column of average reflects the average size of the six geological hazards in the research area. Scales column shows the proportion of various geological hazards on different scales, and the specific three-digit data respectively represent the total number of such disasters on small, medium and large scales. It can be seen that no matter what kind of hazards, the scale of their occurrence is mainly small and medium-sized. In the last column, the formation reason shows the statistical results of the causes of various hazards. The former figure represents the number of





**Table 1.** Detailed inventory of six types of geological hazards in Bijie

| Geological hazards | Total number | Minimum (m2) | Maximum (m2) | Average (m2) | Scales | Formation reasons |
|---|---|---|---|---|---|---|
| Debris flow | 59 | 300 | 300000 | 33540 | 34:21:4 | 58:1 |
| Landslide | 424 | 100 | 9000000 | 65353 | 330:86:8 | 380:44 |
| Collapse | 361 | 2.3 | 5812500 | 11052 | 245:105:11 | 313:48 |
| Unstableslope | 295 | 500 | 1200000 | 104551 | 234:56:5 | 279:16 |
| Groundcollapse | 214 | 0.01 | 500000 | 33752 | 131:64:19 | 89:125 |
| Groundcrack | 82 | 20 | 36000 | 1283 | 50:25:7 | 25:57 |

disasters caused by natural causes, while the latter figure represents the number of disasters caused by human factors. Thus, in addition to the ground collapse and the ground crack, the other four kinds of geological hazards are all formed by natural causes as the main reason. In the study, we will use 70% of the six geological hazards to train various classifiers, and the remaining 30% will be used to verify the classification accuracy of the classifier. What needs to be declared is to make the training results
more accurate,the selection of the set is based on the random principle.

## 4.2 Geological hazards conditioning factors

  Based on the nature of Bijie area, data availability, and quasi-empirical and statistical criteria in the literature, we will choose altitude , aspect ,slope, curvature , plan curvature , profile curvature , stream power index (SPI), topographic wetness index (TWI), sediment transport index (STI),distance to faults , distance to rivers , distance to roads , impact of mining activ-
ities,lithology , landuse and rainfall as the landslide controlling factors (Zhou et al., (2016)). Among them, elevation, slope, aspect, curvature, plane curvature, profile curvature, SPI, TWI, STI, distance to faults, and lithology are all geologically induced factors. Rainfall is a meteorological trigger, while distance to roads, impact of mining activities and landuse are artificial triggers (Bai et al., (2010)).

  Elevation, aspect, and slope have been considered as the most important terrain factors closely related to geological hazards
(Ayalew and Yamagishi, (2005); Chalkias et al., (2016)), In this study, we used the terrain and hydrological Analysis in Arcmap ,basing on 30 m digital elevation model (NASA Shuttle Radar Topography Mission (SRTM) Global 1 arc second product),to extract the slope angle, slope aspect, curvature,plan curvature, profile curvature , SPI ,TWI and STI(Zhao et all., (2018)). The mining activities of coal resources will also cause geological hazards especially landslides, since mining will increase the vertical motion of the ground, which will likely lead to lateral expansion of subsidence. Therefore, disasters are divided into two
categories according to whether they are in the mining area or not. Disasters in the mining area are represented as 1, while those in the non-mining area are represented as 0.The factors of distance to faults, rivers and roads have an important impact on the spread and size of geological hazards in the study area (Pham et al., (2015),Pham et al., (2016)). The data of faults ,rivers and roads data was acquired from Bijie's geographic database,and then were applied by buffer analysis respectively. Moreover, according to the position of the hazards, the values of different grades will be assigned, and the closer the fault and
other factors are, the higher the corresponding values will be.Landuse came from map of land types surveyed in Bijie. Rainfall





**Table 2.** Multicollinearity analysis for the geological hazards conditioning factors

| Geological hazards conditioning factors | Tolerance | VIF |
|---|---|---|
| Altitude | 0.754 | 1.326 |
| Aspect | 0.912 | 1.097 |
| Slope | 0.902 | 1.109 |
| Curvature | 0.341 | 2.933 |
| Plan curvature | 0.046 | 21.739 |
| Profile curvature | 0.037 | 27.027 |
| SPI | 0.869 | 1.150 |
| STI | 0.862 | 1.161 |
| TWI | 0.992 | 1.008 |
| Faults | 0.985 | 1.015 |
| Rainfall | 0.865 | 1.168 |
| Roads | 0.900 | 1.111 |
| Rivers | 0.907 | 1.103 |
| Landuse | 0.964 | 1.038 |
| Lithology | 0.891 | 1.122 |
| Mine | 0.959 | 1.043 |

data was based on the average daily rainfall of 7 counties in the Bijie from 2017 to 2019.Lithology plays a crucial role in the formation of geological hazards,as it can be directly related to the slope stability and different lithology will also affects slope deformation (Guo et al., (2015); Saha et al., (2002)).The lithology in this research is based on the lithology map in Bijie.

## 5    Results and analysis

### 5.1    Geological hazards conditioning factors analysis

In this study, multicollinearity among the conditioning factors were identified using the tolerances and variance inflation factors (VIF) (Table 2) and Pearson's correlation coefficient (Fig.6).The results show that the profile curvature and the plane curvature have smaller tolerances (0.037, 0.046) and larger VIF values (27.027, 21.739). Neither the variance of the two condition factors nor the VIF values meet the threshold values (VIF <10 or Tolerance> 0.1). In the case of Pearson's correlation,

the maximum correlation value occurs between curvature and profile curvature, and the correlation value of the two is -0.877. This is followed by a correlation of curvature and plane curvature of 0.832. At the same time, the two values are greater than 0.7, indicating that the curvature, profile curvature and plane curvature are directly strongly collinearity. Therefore, through the above analysis, we need to remove the plane curvature and profile curvature from the prediction processes.





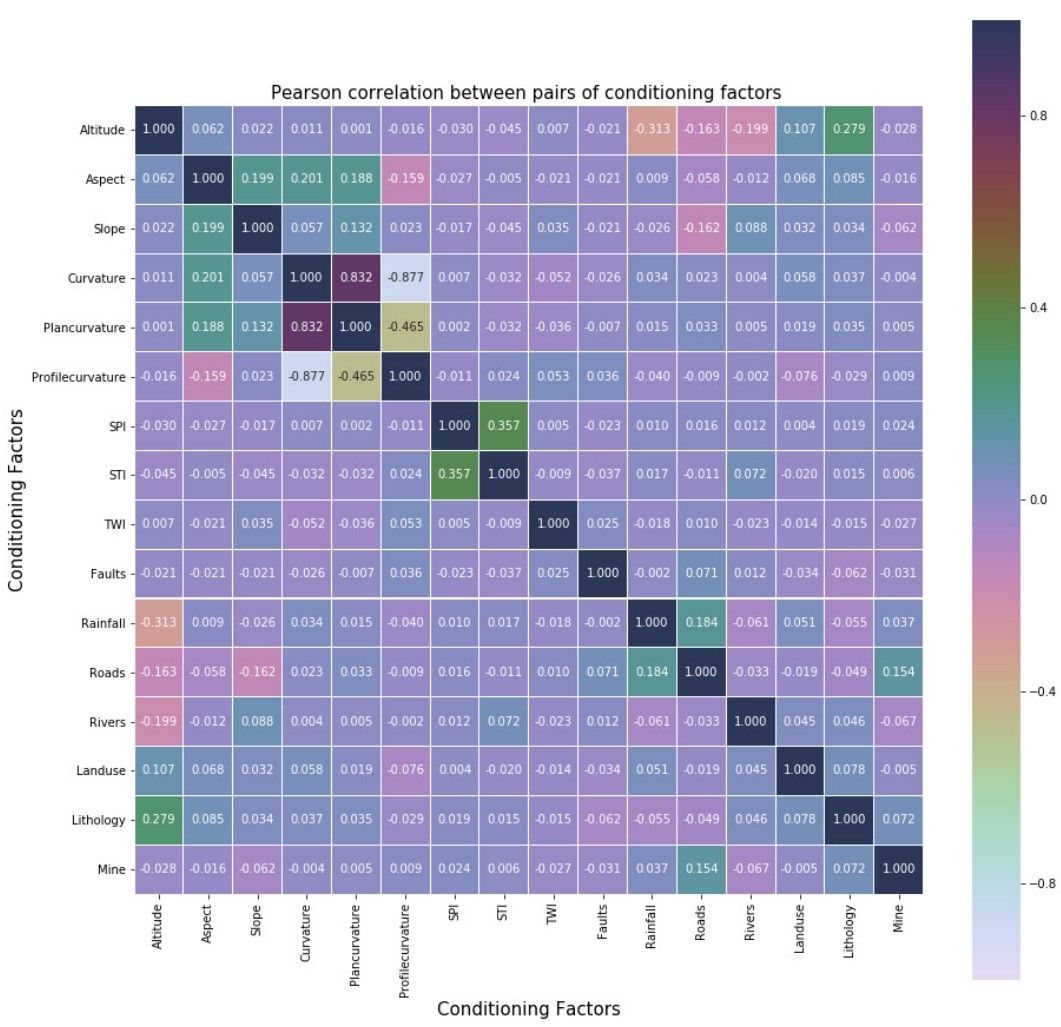

**Figure 6.** Pearson correlation between pairs of geological conditioning factors

## 5.2 Parameters for classification models

In this section, we will use the 10 classification models introduced in Sect. 3 for the classification and modeling of 6 types of geological hazards in Bijie area. The source code of all the classification methods mentioned earlier is implemented in



**Table 3.** The parameters of classification models

| Models | Parameters | Note |
|---|---|---|
| LR | r= L2 ; f= Newton's method; i=100 | r is the regularization used in the penalization; f is the loss fuction;i is the maximum number of iterations |
| LDA | s=SVD | s is the algorithm for solving the optimization problem |
| NBC | p= MLE | p is the method of calculating the prior probability |
| MLP | u1= 80,u2= 30,u3=10; a=ReLU; f=SGD | u1,u2, u3 are the number of neurons in the first, second and third hidden layers, respectively; a is the activation function |
| SVM | c=20;g=1.5 | c is the penalty factor; g is the parameter of the kernel function |
| DT | d=5;l=3 | d is the maximum depth of the tree; l is the minimum number of training samples for each child node |
| KNN | k=5 | k is the number of neighboring nodes |
| Adaboost | i=50,r=0.5 | r is the learning rate |
| GBDT | | |
| RF | i=100 | |

the jupyter environment based on the Python language combined with the open source framework Scikit-learn. Scikit-learn was launched by data scientist David Cournapeau in 2007 and requires support from other packages such as NumPy and SciPy. It is an open source framework developed in Python for machine learning applications. The results were obtained using a PC equipped with an Intel Core i7-8550U processor.As stated in Sect. 5.1.2, fourteen important factors, namely altitude, aspect, slope,curvature, SPI,STI,TWI, lithology,rainfall, landuse,mine or not, distance to rivers, faults and roads were selected to establish the classification models. In this study, the parameters of various classifiers are shown in Table 3. In LR modeling, L2 regularization and Newton method were used as loss function training samples, and the logistic regression equations of 6 types of geological hazards was obtained as follows. LDA used singular value decomposition(SVD) to solve the optimization algorithm. Gaussian naive bayes algorithm is selected in NBC model and the prior probability of sample data is calculated by maximum likelihood estimation(MLE). In the MLP model, ReLU was selected as the activation function, and random gradient descent with minbatch equal to 64 was selected for algorithm optimization in the training. The network structure was set to include 3 hidden layers. The SVM kernel function is set to the gaussian kernel function with a coefficient of 1.5, and the penalty coefficient of the objective function is set to 20. In the decision tree model, information entropy is used to calculate the impurity of nodes. In order to solve the over-fitting problem caused by the impurity of optimization, the maximum depth of the decision tree is limited to 5, and all the child nodes must contain at least 3 training samples, otherwise branching will not





occur. The KNN model sets the number of adjacent nodes to 5. In Adaboost and GBDT models, in order to prevent overfitting, the maximum iteration number is set to 50, the learning rate is 0.5, and the maximum iteration number is set to 100 in RF.

$$
\begin{aligned}
Collapse = -6.189 &- 0.682 * Altitude - 1.252 * Aspect + 3.939 * Slope - 1.389 * Curvature \\
&+ 2.172 * SPI + 1.53 * STI + 3.966 * TWI + 2.755 * Faults + 6.533 * Rainfall \\
&- 1.486 * Roads - 2.402 * River + 5.125 * Landuse - 0.952 * Lithology + 1.753 * Mine
\end{aligned} \tag{10}
$$

$$
\begin{aligned}
Groundcrack = -0.39 &- 0.671 * Altitude + 3.553 * Aspect - 3.773 * Slope - 1.472 * Curvature \\
&- 0.143 * SPI - 3.228 * STI - 2.375 * TWI + 0.759 * Faults + 1.422 * Rainfall \\
&- 3.56 * Roads - 3.718 * River - 1.039 * Landuse + 9.433 * Lithology + 8.494 * Mine
\end{aligned} \tag{11}
$$

$$
\begin{aligned}
Groundcollapse = -0.103 &- 1.11 * Altitude + 4.937 * Aspect - 8.253 * Slope + 4.779 * Curvature \\
&- 0.35 * SPI - 4.141 * STI + 2.533 * TWI - 1.207 * Faults - 6.204 * Rainfall \\
&+ 1.158 * Roads - 0.66 * River + 2.413 * Landuse - 5.529 * Lithology + 8.045 * Mine
\end{aligned} \tag{12}
$$

$$
\begin{aligned}
Landslide = -5.263 &- 2.014 * Altitude + 8.041 * Aspect - 1.318 * Slope + 3.585 * Curvature \\
&- 0.125 * SPI + 3.732 * STI + 7.117 * TWI - 2.654 * Faults + 5.772 * Rainfall \\
&+ 1.158 * Roads + 1.053 * River - 4.679 * Landuse - 2.723 * Lithology - 3.808 * Mine
\end{aligned} \tag{13}
$$

$$
\begin{aligned}
Debrisflow = -2.132 &+ 2.97 * Altitude - 9.843 * Aspect - 1.883 * Slope - 3.619 * Curvature \\
&+ 0.247 * SPI - 7.968 * STI - 3.856 * TWI - 1.403 * Faults - 1.046 * Rainfall \\
&- 4.63 * Roads + 1.046 * River - 6.562 * Landuse - 2.029 * Lithology - 8.12 * Mine
\end{aligned} \tag{14}
$$

$$
\begin{aligned}
Unstableslope = -2.526 &+ 0.793 * Altitude - 1.438 * Aspect + 2.448 * Slope - 1.03 * Curvature \\
&- 0.193 * SPI - 1.978 * STI - 4.006 * TWI + 1.38 * Faults - 1.431 * Rainfall \\
&+ 7.823 * Roads - 4.722 * River + 2.386 * Landuse + 1.721 * Lithology - 1.714 * Mine
\end{aligned} \tag{15}
$$

## 5.3 Model validation and comparison

The receiver operating characteristic(ROC) curve (Hanley and McNeil, (1983)) is cutoff-indepedent, allowing intermediate states to exist. The Area under Curve (AUC) refers to the area under the ROC curve, which can be used to intuitively evaluate the





performance of the classifier. Its value is between 0.1 and 1, and a classifier with a larger AUC value is considered better.Using
the validation data set, ROC curve, AUC evaluated the performance of 10 classifier models and the results of ROC curve and

AUC are shown in Fig.7. It can be seen that all models have good classification ability, and the ROC curve basically shows a
steep trend, that is, the higher the TPR, the lower the FPR. This is because TPR represents the coverage of the model's predicted
response, while FPR represents the model's false report .The degree of response of the ROC curve model is therefore better
as the steeper. At the same time, the average AUC values are all above 0.8, and the classification ability of the SVM model
is the strongest (AUC = 0.941), indicating that the classification results of the six types of geological disasters are basically

consistent with the actual observations. In the classification and assessment of different geological hazards, the MLP model
has the highest prediction ability for ground collapse (AUC = 0.969), and the worst model is the performance of decision
trees in unstable slopes (AUC = 0.704). In the SVM and MLP models, the SVM uses the principle based on structured risk
minimization, while the MLP model uses the traditional empirical risk minimization principle. SVM based on structural risk
minimization achieves the highest classification prediction accuracy, and its solution has global optimality, while MLP usually

results in a local optimal solution.

Fig.8 shows the precision and sensitivity of 10 models for each type of geological hazard. It can be seen from the figure that,
excluding debris flow and unstable slope, SVM has the highest probability of correctly classifying pixels as the corresponding
geological disaster category, specifically: collapse (0.938), ground collapse (0.767), ground collapse (0.967) and landslide
(0.956). It's not hard to find that the precision of SVM for ground collapse is the largest among the six types of geological

disasters. For the remaining two geological disasters, RF has the highest probability of correctly classifying pixels as debris
flows(0.735), and the highest precision rate of unstable slopes(0.880) belongs to the GBDT model. As for sensitivity ,SVM has
the highest valuesto collapse (0.870), ground collapse (0.897), debris flow (0.917), and unstable slopes (0.900), indicating that
the pixels of these four geological disasters in the SVM model have been classified correctly for the corresponding categories
of high. The MLP neural network model and GBDT model have the highest sensitivity to ground collapse (0.870) and debris

flow (0.875), respectively.

As a multi-classification problem, we also used common multi-class evaluation indicators which includes Kappa index,
Macro F1, Micro F1 to evaluate the classification models,the results are shown in Table 4 below. The value of Kappa index of
all models ranges from 0.560 to 0.845, indicating that the prediction results and observation results are basically consistent.
Usually, Macro F1 and Micro F1 are used to evaluate the average performance of the whole classification and classification

with high both values works well (Liu C et al. ,2017). In all models, SVM has both the highest Macro F1 and Micro F1 values,
which is consistent with the results reflected by the AUC value.



**Figure 7.** The ROC curves of the ten models:a LR,b:LDA,c:NB,d:MLP,e:SVM,f:DT,g:KNN,h:AdaBoost,i:GDBT,j:RF




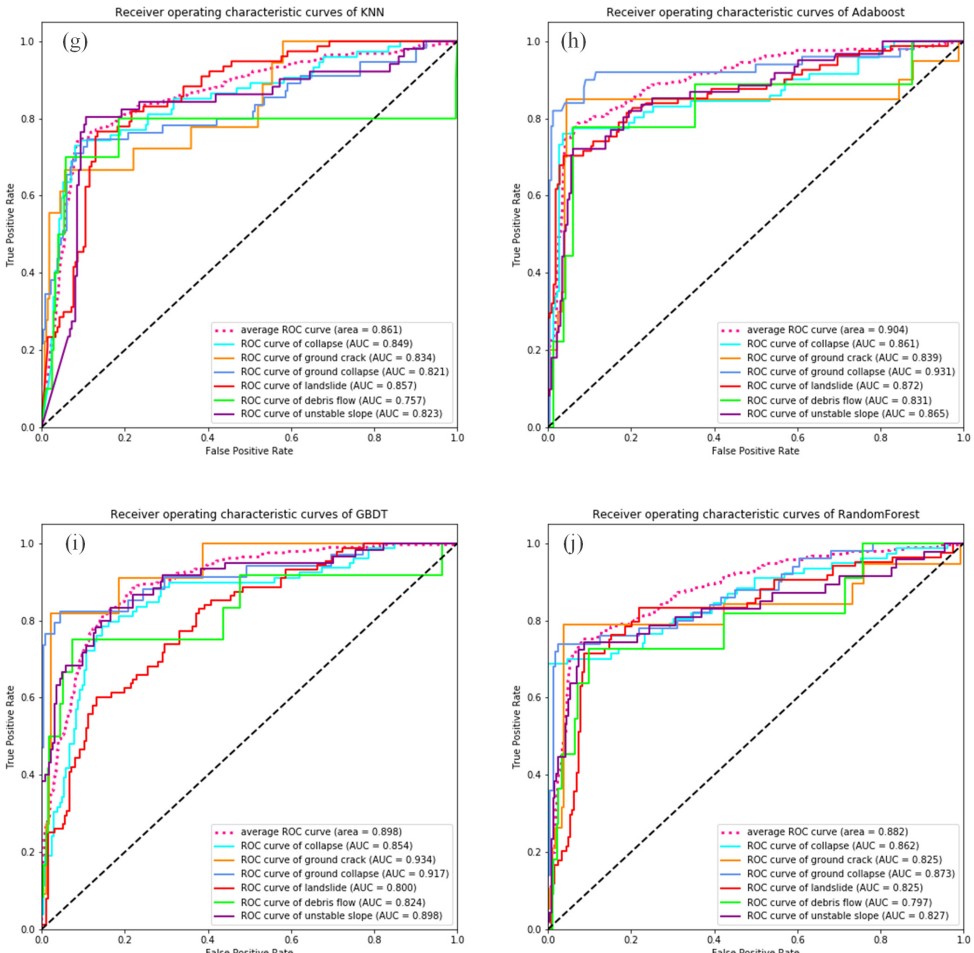

**Figure 8.** (continued)


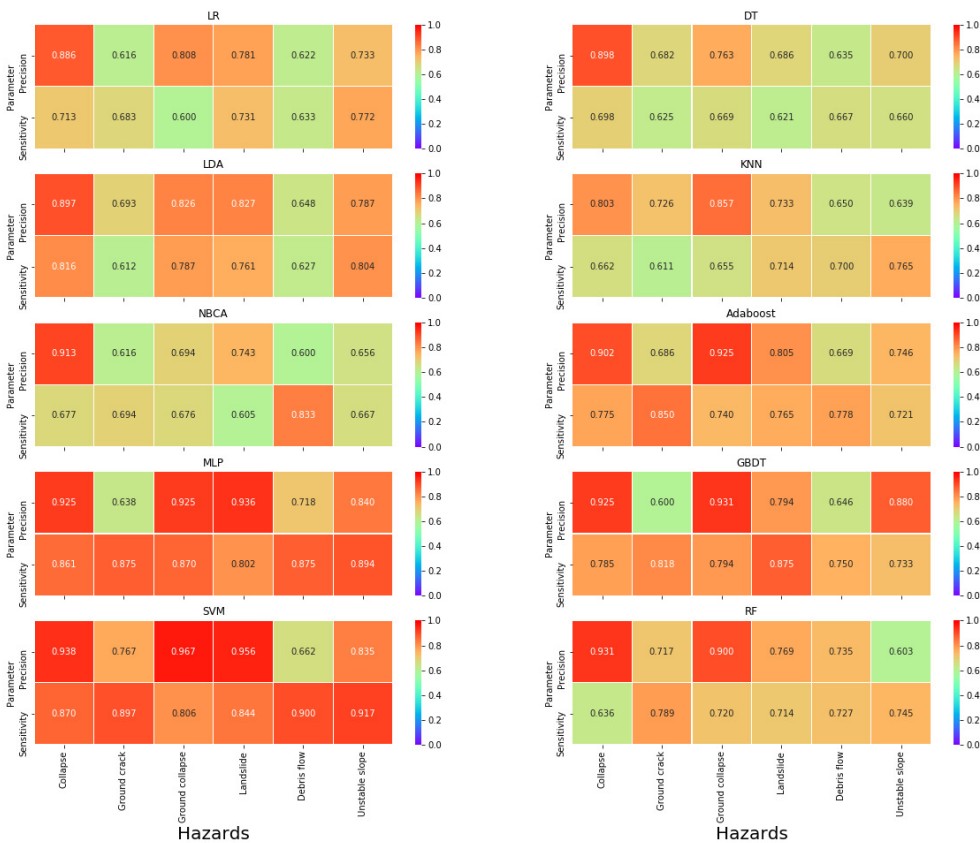

**Figure 9.** Precision and sensitivity of ten classification models

Finally, the 5% significance level Friedman test was used to perform statistical nonparametric significance tests on the results of all classification models. The results showed that the p-value (0.269) was greater than 0.05. Therefore, the original hypothesis was accepted and the classification results of the model were not significant difference.

### 5.4 Variable importance

Fig.9 shows the results of the assessment of variable importance. It can be seen that all the conditioning factors in the six geological hazards have contributed to the models. However, conditioning factors behave differently in different geological hazards and in different models. For example, in collapse, altitude and slope are important factors, while in ground collapse, mineral activity is the important factor. At the same time, the contribution rate of other condition factors such as mineral activity to the collapse will be varied depending on the models. In the LDA model, the contribution rate of mineral activity is the smallest one (0.025). At the same time, its contribution rate is second (0.138) only to altitude (0.164) and slope (0.141) in




**Table 4.** Models validation

| Models | Kappa index | Macro F1 | Micro F1 |
|---|---|---|---|
| LR | 0.651 | 0.660 | 0.724 |
| LDA | 0.668 | 0.658 | 0.740 |
| NBC | 0.560 | 0.601 | 0.652 |
| MLP | 0.806 | 0.777 | 0.849 |
| SVM | 0.845 | 0.851 | 0.878 |
| DT | 0.544 | 0.607 | 0.638 |
| KNN | 0.613 | 0.642 | 0.691 |
| Adaboost | 0.701 | 0.711 | 0.760 |
| GBDT | 0.744 | 0.751 | 0.803 |
| RF | 0.634 | 0.661 | 0.705 |

**Table 5.** Result of Friedman test for the ten models with $\alpha$=0.05

| Value in$\chi^2$ | p value |
|---|---|
| 11.096 | 0.269 |

the SVM model.

Moreover, we can also catch sight that altitude, slope, and rainfall have a larger contribution rate in any model of landslides, debris flows, and unstable slopes. According to the actual situation, as far as landslides are concerned, 94.4% of the landslides
in Bijie City occures at altitudes higher than 1000m. The average slope of all landslide sites is 15.542 °, and landslides affected by rainfall account for 87.5% of the total landslides. In ground crack and ground collapse, man-made mineral activities have the largest contribution rate. It can also be obtained from Table 1 that in these two types of geological hazards, the number of disasters caused by human factors is greater than the number of disasters caused by natural factors.

Natural Hazards
and Earth System


**Figure 10.** Importance of conditioning factors to the ten models





## 6   Conclusions

Located in the southwestern plateau of China, Bijie City, where karst landforms are widely distributed, belongs to a high incidence of geological hazards. Debris flows, landslides, unstable slopes, ground crack, ground collapse and collapse are the major geological hazards in Bijie city. Using conditioning factors to construct different classification models, analyzing and comparing the performance capabilities of the models, and studying the nonlinear relationships between conditioning factors and geographical disasters are the focus of this study. In this article, we evaluated and compared a total of ten classification

models, including discriminant analysis and logistic regression in traditional classification methods, as well as many new classification machine learning methods and techniques (such as Adaboost, GBDT and random forest algorithms in ensemble learning) ,and some of the most popular methods (such as MLP neural network models and SVM).

In general, in terms of the classification results of the six types of geological hazards, the average AUC values under the ROC curves of the ten classification models selected in this study are all greater than 0.8 (Fig.7). It can also be found that SVM

is significantly better than other models in terms of overall (0.941) and classification results of individual geological hazards (collapse: 0.949, ground crack: 0.907, ground collapse: 0.952, landslide: 0.830, displacement flow: 0.963, slope: 0.922) .It can be proved that these ten classification models all show good performance in this study. However, in some cases, using AUC to evaluate model performance may not be the best method, because a high value of AUC may not guarantee high spatial accuracy of the model (Aguirre-Gutiérrez et al., (2013)). Therefore, this study also provided precision and sensitivity assessment mea-

sures (Fig.8). It can be seen that the values of precison and sensitivity are between 0.603-0.967 and 0.600-0.917, respectively. At this time, although SVM still has the most maximum value, the highest accuracy and sensitivity of different geological disasters are no longer completely focused on the same model. The highest precision value of debris flow(0.735) belongs to RF model, and the highest sensitivity value of ground collapse(0.870) belongs to MLP neural network model.

Kappa index, Macro F1 and Micro F1 are also used to evaluate these classification models. Table 4 shows that values of

Kappa index of these ten models range from 0.560 to 0.845, indicating that the predicted results are basically consistent with the observed results. In all models, SVM still has the highest Macro F1(0.851) and Micro F1(0.878) values, which is consistent with the results reflected by AUC values. Finally, the Friedman test of 5% significance level was used to conduct statistical nonparametric significance test on the results of all classification models. The statistical results showed that the p value (0.269) was greater than 0.05, that is, there was no significant difference in the classification results of the model.

The selection of conditioning factors and the determination of categories are key factors affecting the quality of models (Irigaray et al., (2007); Costanzo et al., (2012)), although for different geological hazards have some selection criteria factor method is proposed, such as GIS matrix (Cross, (1998)) Cross combination method and linear correlation (Irigaray et al., (2007)), the final choice of conditioning still need to combine the nature of the study area, data availability, literature and statistical standards (Zhou et al., (2016)). In general, topography, geology, hydrology, meteorology, and the impact of human

activities have been widely used as geological hazard regulators.

When establishing the classification model, some of the conditioning factors in the initial data set will not bring good prediction ability to the model, but will generate noise and thus reduce the performance of the model. Therefore, feature selection



is needed before establishing the model (Martínez-Álvarez et al., (2013)). In this study, we used VIF, tolerance, and Pearson's correlation coefficient method to detect multicollinearity between the conditioning factors in the initial data set (Table2 and

Fig.6). The results showed that the tolerance and VIF values of profile curvature and plane curvature did not reach the threshold value, and in Pearson's correlation coefficient method, the curvature showed strong collinearity with profile curvature and plane curvature. Therefore, through feature selection, we will remove the plane curvature and profile curvature from the classification model. After the construction of the classification model, we evaluated the contribution of the conditioning factors to different geological hazards and different models (Fig.9). We find that the same conditioning factors have disparate meanings for differ-

ent geological disasters. For example, altitude and slope are the most important factors in collapse, but are far less important in surface crack than mining activities. In the meantime, the same conditioning factor for the identical geological hazard will have diverse meanings in different models. For example, excluding altitude and slope, the importance of other factors to collapse will be changed depending on the types of models used. In the LDA model, the contribution rate of mining activities is only 0.025. Whereas, its contribution rate is the third most importance after elevation (0.164) and slope (0.131) in the SVM model.

Apart from the differences, we can also find some similarities. The contribution rates of elevation, slope and rainfall are higher in landslide, debris flow and unstable slope. This is reasonable considering that more than 90% of the landslides (94.4%), debris flows (98.3%) and unstable slopes (96.3%) in Bijie all occur above 1000m. The average slope at the occurrence of landslide is 15.542°, debris flow is 15.421°, and unstable slope is 17.713°. As for rainfall effects, 87.5% of landslides, 66.1% of debris flows and 63.1% of unstable slopes will be affected by rainfall. For ground collapse and ground crack, the disas-

ters caused by human factors are far more than those caused by natural factors (Table 1), which can be explained that the contribution rate of mining activities to both is the largest in all models.

Dataavailability The one-arc-second SRTM DEM is downloaded from the website http://e4ftl01.cr.usgs621.gov /MODV6_Dal_D/SRTM/SRTMGL1.003/2000.02.11/

Three-arc-second SRTM DEM is downloaded from http://www2.jpl.nasa.gov/srtm/cbanddataproducts.html

*Author contributions.*  The main concept, draft, methodology, writing and revisions were done by JQS. JZ contributed to the methodology and revisions and supported the work via proofreading. CYS was mainly involved in data processing and post revision of articles .

*Competing interests.*  The authors declare that they have no conflict of interest.

*Acknowledgements.*  The authors are very thankful to The Third Surveying and Mapping Institute of Guizhou Province, Guiyang,China for its data support.



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
