# Peer review of "Construct and evaluate the classification models of six types of geological hazards in Bijie city, Guizhou province, China"

_Natural Hazards and Earth System Sciences, 2020_

## Referee Comment (RC1) · Anonymous Referee #1 · 25 May 2020

The paper is not acceptable for the review: the English form is very bad and some of the geological hazards are unclear. The writer stopped the review at the end of the introduction.

Other spotted errors and comments are as follows:

Line 1 "ground collapse, collapse and ground collapse" ?????????????????

Line 5: "manual field survey"?????

Line 6 place "the" before "inventory"

Line 7 ratio or percentage?

[Figure]

Lines 9-12 "In order to select the optimal subset of the conditioning factors, the multicollinearity of these factors was assessed using tolerances and variance inflation factors(VIF) and Pearson's correlation coefficient, and factors with multicollinearity were excluded to optimize the model. Subsequently, ten classification models were structured, and the models were verified and compared by using the receiver operating characteristic(ROC), precision, sensitivity, Kappa coefficient and F1 values.I" Unclear sentences

Lines 15-18 "Among them, the average AUC value(0.941), AUC values for individual geological hazards (collapse: 0.949, ground crack: 0.907, ground collapse: 0.952, landslide: 0.830, displacement flow: 0.963, slope: 0.922),Kappa coefficient (0.845), Macro F1(0.851) and Micro F1(0.878) of SVM all had the highest values." Very unclear period in which different quantities are mixed together.

Lines 20-24 "Most cities in China are located in regions that are extremely vulnerable to a wide range of natural hazards, particularly geological hazards, and the assessments for hazardousness, vulnerability and risk of natural hazards for China shows that on the whole, high hazardousness regions concentrate in western (Wang et al., (2008), Liu et al., (2012),Zhuang et al., (2016)). Guizhou province, located in the plateau region of southwest China, not only has extensive carbonate rock distribution (70%) and karst development; In addition, due to the large crustal uplift and severe deformation recently, as well as the induction. . . . . . . . ." Very unclear period

Lines 25-26 "Among them, debris flow, landslide, unstable slope, ground collapse, collapse and ground crack are the main geological disasters in Guizhou." What are ground collapse? And collapse? Perhaps the author mean the land subsidence or something else?

Line 28 The references of Cannon et al. 2007-2010 and Staley et al. 2017 concern post-wild-fires debris flows, a very particular category that in China is rare. The writer supposes that debris flows occurring in Guizhou province be runoff-generated debris

flows (Imaizumi et la, 2006; Gregoretti and Dalla Fontana, 2008; Kean et al. 2013) as those occurring in other parts of China (see Ma et al., 2018; Chen et al, 2019; Liu and He, 2020)

Lines 28-30 "And the debris flows in Guizhou province are mainly distributed in the western part of the province , ranging from several hundred thousand to several million in size." Size of what?

What do the authors mean for landslide and collapse?

Lines 35-36 ".As for ground collapses and cracks, most of the ground collapses areas that have been found in the province occur in carbonate areas." Unclear sentence

Lines 45-60: the introduction of the learning machine methods is too long, redundant and confused. Moreover, before introducing machine learning method, it must clearly explained the reasons of using them.

Lines 60-64. The disaster conditioning factors are introduced without any explanation about their use and their possible links with the machine learning method.

The writer suggests the authors to re-write the paper, better explaining the phenomen, linking the factors to the physics of debris flow occurrence and widening the discussion of results.

Chen, M., Liu, X., Wang, X., Zao, T., Zhou, J., 2019. Contribution of excessive supply of solid material to a runoff-generated debris flow during its routing along a gully and its impact on the downstream village with blockage effects. Water 11, 169. https://doi.org/10.3390/w11010169.

Gregoretti, C., Dalla Fontana, G., 2008. The triggering of debris flow due to channel-bed failure in some alpine headwater basins of the Dolomites: analyses of critical runoff. Hydrol. Process. 22, 2248–2263. https://doi.org/10.1002/hyp.6821.

Imaizumi, F., Sidle, R.C., Tsuchiya, S., Ohsaka, O., 2006. Hydrogeomorphic processes in a steep debris ïñĆow initiation zone. Geophys. Res. Lett. 33, L10404. https://doi.org/10.1029/2006GL026250.

Kean, J.W., McCoy, S.W., Tucker, G.E., Staley, D.M., Coe, J.A., 2013. Runoff-generated debris ïñĆows: observations and modeling of surge initiation, magnitude and frequency. J. Geophys. Res. 118, 2190–2207. https://doi.org/10.1029/jgrf20148.

Liu W, He S. 2020 Comprensive modelling of runoff-generated debris ïñĆows from formation to propagation in a catchment. Landslide https://doi.org/10.1007/s10346-020-01383-w

Ma, C., Deng, J., Wang, R., 2018. Analysis of the triggering conditions and erosion of a runoff triggered debris ïñĆow in Miyun County, Beijing, China. Landslide https://doi.org/10.1007/s10346-018-1080-3.

---

## Referee Comment (RC2) · Anonymous Referee #2 · 5 Jul 2020

The paper entitled "Construct and evaluate the classification models of six types of geological hazards in Bijie city, Guizhou province,China" (please note the missing space after the comma) presents a study case of something that might be associated with a susceptibility assessment and somehow with a multi-hazard approach. Unfortunately, while such a study case might be interesting, there is nothing in the manuscript to represent a substantial contribution to the understanding of natural hazards and their consequences. More, the language is not scientific and the English language is very poor (for example in the first phrase there are missing articles; usually we speak about landslides and not landslide), even with too many misspelling errors and missing spaces, a situation that makes it very hard to read (spaces are missing, commas appear where

is not the case and the proper punctuation is not used). Often, even the conceptual basics of natural hazards and their modeling are missing: - from the first phrase hazard, vulnerability and risk are not well introduced; - the usual international natural hazard and risk terminology is also poor included, disasters being considered as hazards. The natural hazards are not well defined, for example: - the so-called collapse and ground collapse might seem different phenomenon, but are not referenced in the literature; - debris flows are not considered a category of landslides; - natural and human-induced processes are mixed. Also, the modeling approach speaks about classification, when actually such a study case needs a probabilistic approach. The chosen factors are not related to natural hazard processes. I have given a lot of thinking after reading multiple times the article, in order to try to give it some directions toward a natural hazards approach, but unfortunately, the shortcomings of the paper are too many. A full reconsideration of the problem is needed. The authors should choose a single natural hazard, and try to map it (there is no description of the inventory and on the methodology) and then probabilistically model it. The literature review is poor, many fundamental papers are missing. I do not see how could this paper reach a level for publishing without reconsidering every aspect. The English language needs a professional touch for sure also.